# “Omics” and Postmortem Interval Estimation: A Systematic Review

**DOI:** 10.3390/ijms26031034

**Published:** 2025-01-25

**Authors:** Laura Secco, Stefano Palumbi, Pasquale Padalino, Eva Grosso, Matteo Perilli, Matteo Casonato, Giovanni Cecchetto, Guido Viel

**Affiliations:** 1Legal Medicine and Toxicology, Department of Cardiac, Thoracic, Vascular Sciences and Public Health, University of Padova, Via G. Falloppio 50, 35121 Padova, Italy; laura.secco@studenti.unipd.it (L.S.); stefano.palumbi@studenti.unipd.it (S.P.); pasquale.padalino@studenti.unipd.it (P.P.); eva.grosso@studenti.unipd.it (E.G.); matteo.perilli@studenti.unipd.it (M.P.); matteo.casonato@studenti.unipd.it (M.C.); 2Legal Medicine, Department of Public Health, Experimental and Forensic Medicine, University of Pavia, 27100 Pavia, Italy; cecchetto.gio@gmail.com

**Keywords:** postmortem interval, time-since-death, mass-spectrometry, forensic, omics, proteomics, metabolomics, lipidomics

## Abstract

Postmortem interval (PMI) estimation is a challenge of utmost importance in forensic daily practice. Traditional methods face limitations in accuracy and reliability, particularly for advanced decomposition stages. Recent advances in “omics” sciences, providing a holistic view of postmortem biochemical changes, offer promising avenues for overcoming these challenges. This systematic review aims at investigating the role of mass-spectrometry-based “omics” approaches in PMI estimation to elucidate molecular mechanisms underlying predictable time-dependent biochemical alterations occurring after death. A systematic search was performed, adhering to PRISMA guidelines, through “free-text” protocols in the databases PubMed, SCOPUS and Web of Science. The inclusion criteria were as follows: experimental studies analyzing, as investigated samples, animal or human corpses in toto or in parts and estimating PMI through MS-based untargeted omics approaches, with full texts in the English language. Quality assessment was performed using STROBE and ARRIVE critical appraisal checklists. A total of 1152 papers were screened and 26 included. Seventeen papers adopted a proteomic approach (65.4%), nine focused on metabolomics (34.6%) and two on lipidomics (7.7%). Most papers (57.7%) focused on short PMIs (<7 days), the remaining papers explored medium (7–120 days) (30.77%) and long PMIs (>120 days) (15.4%). Muscle tissue was the most frequently analyzed substrate (34.6% of papers), followed by liver (19.2%), bones (15.4%), cardiac blood and leaking fluids (11.5%), lung, kidney and serum (7.7%), and spleen, vitreous humor and heart (3.8%). Predictable time-dependent degradation patterns of macromolecules in different biological substrates have been discussed, with special attention to molecular insights into postmortem biochemical changes.

## 1. Introduction

The estimation of the postmortem interval (PMI), defined as the time elapsed since death, represents a complex and pivotal challenge for forensic pathologists in daily practice. The role of PMI becomes crucial in the context of criminal investigations. Indeed, it may direct suspicions towards a certain perpetrator, defining the whereabouts and last movements of the victim and corroborating or rebutting witness testimony in court [1,2,3].

Throughout the decades many different approaches have been developed and proposed by forensic pathologists for PMI estimation. In forensic daily casework the most widely used methods rely on the assessment of early postmortem changes. These are represented by *algor*, *livor*, and *rigor mortis*, and the gross visual evaluation of transformative destructive and special phenomena, physiologically affecting corpses during early and advanced postmortem phases, respectively [4,5,6].

Such approaches are often complemented and assisted by forensic entomology analyses, which provide useful information for PMI inference through the examination of types and life cycles of insects commonly colonizing cadavers [7,8]. Other analytical methods, although less widely applied, comprehend the study of supravital reactions (related to residual electrical or pharmacological excitability of body tissues) [9,10], the quantitation of vitreous potassium concentrations [11,12], and histology investigation of different tissue substrates [13,14,15,16,17].

Nevertheless, the aforementioned conventional methods are highly subjective, limited to specific postmortem phases, and strongly influenced by a wide range of variables, such as extrinsic environmental factors (i.e., temperature, humidity, ventilation, the action of micro- and macro-fauna on the body) and intrinsic factors (i.e., sex, age, cause of death, underlying pathologies, body mass index). These variables impact postmortem changes and corpse decomposition processes, potentially leading to unreliable PMI estimations. Moreover, for most of the above traditional methods, accuracy and precision decrease as decomposition advances, demonstrating poor feasibility in cases of longer PMIs [1,3,18,19].

To overcome these drawbacks, recent scientific research has focused on developing new biochemical and molecular approaches for PMI estimation, mainly based on the examination of postmortem biochemical degradation patterns. These include hypoxia-induced changes such as altered enzymatic reactions, cellular autolysis, and cessation of anabolic pathways, as well as microbial breakdown processes [20,21,22,23].

Omics technologies aim at reconstructing the biology of the investigated system (i.e., the cell, tissue, or organism) through the detection of genes (genomics), mRNA (transcriptomics), proteins (proteomics), and metabolites (metabolomics) in a specific biologic sample in a holistic, non-targeted and non-biased manner. These approaches have recently gained increasing importance also in forensic practice, as they provide a comprehensive analysis of postmortem alterations at the molecular level in different substrates through high-throughput detection techniques. Therefore, they offer great potential for more accurate and objective time since death estimation [24,25,26,27,28,29,30]. Proteomic approaches proved to be more reliable than DNA and RNA degradation-based analysis, as protein markers are more resistant, less susceptible to extrinsic factors, and exhibit slower and more reproducible degradation patterns than nucleic acids [3,6]. Moreover, due to their postmortem stability, proteins are more suitable for long-term PMI estimation compared to metabolites and lipids, which are better markers for shorter PMIs.

Therefore, the application of an integrated multi-omics approach combining proteomics, metabolomics, and lipidomics (defined as the “Forensomics approach”), has been recommended, allowing a wider range of PMIs to be investigated [23].

Mass-spectrometry (MS) is one of the leading analytical tools used for “omics” approaches, enabling simultaneous identification and quantitation of a large number of endogenous and exogenous low molecular weight biomolecules, according to mass-to-charge ratio (*m*/*z*) of ions; this is achieved through a multi-stage analytical process based on ionization of molecular compounds, separation of ionized molecules through MS coupling devices (i.e., capillary electrophoresis, gas chromatography, liquid chromatography) and subsequent mass spectrometric detection and analysis [31,32].

MS-based “omics” approaches are currently promising and powerful analytical tools in terms of time since death estimation. They provide a large-scale analysis of molecular compounds and degradation by-products in multicellular systems with high accuracy, sensitivity, and reproducibility [33,34]. Indeed, many recent experimental studies have focused on investigating, through spectrometric techniques, postmortem changes in levels of specific proteins and metabolites in different biological tissues and fluids, finding out predictable time-dependent alterations after death and thus highlighting their role as novel potential biomarkers of PMI [2,3,17,19,21,22,35,36].

In this review, we aimed to investigate the current state-of-the-art regarding the role of mass spectrometry-based “omics” approaches in the analysis of postmortem degradation patterns of biomolecules, offering a general overview of the most important molecular candidates for postmortem interval estimation. Special attention has been paid to elucidating molecular mechanisms underlying the intricate biochemical time-related changes occurring after death.

## 2. Methods

This systematic review was carried out following the criteria included in the 2020 Preferred Reporting Items for Systematic Reviews and Meta-Analyses (PRISMA) guide [37]. In February 2024, two authors (LS, SP) performed a systematic literature search via “free-text” protocols in the PubMed, SCOPUS, and Web of Science databases, with time limits from 1 January 2003 to 1 February 2024. Search terms used for the three databases were as follows: (PMI OR “postmortem interval” OR “post-mortem interval” OR “post mortem interval” OR “time after death” OR “time since death” OR “time of death”) AND (forensic* OR medicolegal OR “legal medicine” OR legal OR “medico-legal”) AND (omic* OR genomic* OR metabolomic* OR proteomic* OR molecular OR transcriptomic* OR lipidomic* OR DNA OR RNA OR mRNA OR miRNA* OR microRNA* OR protein* OR proteic OR “fatty acid*” OR microbiome OR microbiology).

In the Scopus database, the following adjunctive filters for the “subject area” were used: “Medicine”, “Biochemistry, Genetics and Molecular Biology”, “Immunology and Microbiology”, “Agricultural and Biological Sciences”, “Chemistry”, “Nursing”, “Multidisciplinary”, “Veterinary”, “Pharmacology, Toxicology and Pharmaceutics”, “Neuroscience”, “Dentistry” and “Health Professions”.

In both Scopus and Web of Science, the query was launched by selecting only manuscripts available in English.

### 2.1. Paper Selection

Paper selection was conducted by three authors (GV, LS, and SP), based on titles and abstracts of the results retrieved by the systematic search. The following inclusion and exclusion criteria were adopted. Any discrepancies in the paper selection and data extraction were settled by consensus discussion.

#### 2.1.1. Inclusion Criteria

The following inclusion criteria were applied.Titles and abstracts available in the English language.Experimental studies including, as investigated samples, animal or human corpses in toto or in parts (i.e., organs, tissues, and/or fluids) aiming at estimating PMI.Experimental studies estimating PMI through mass-spectrometry-based untargeted omic approaches.

Only studies matching all of the aforementioned criteria were included.

#### 2.1.2. Exclusion Criteria

Letters to the editor, book chapters, reviews, conference proceedings.Full texts not available in the English language.Studies estimating PMI through in vitro experiments.Records estimating PMI through the analysis of the postmortem colonizing fauna.

Not meeting all of the inclusion criteria (A, B, C) or, conversely, meeting at least one of the exclusion criteria (D, E, F, G) was a reason for paper exclusion.

Data extraction from the included records was performed independently by four authors (LS, SP, MP, and MC), who also created a database with the extracted data. The following items were collected from each study: first author, year and journal of publication, best JCR quartile (2024), sample size and type (A for animal, H for human), time interval after death explored, analytical technique and statistical approach applied, selected biomarkers, presence of a PMI predictive model, diagnostic efficiency, and error interval/accuracy of selected markers.

Any discrepancies in the data extraction process were settled by consensus discussion performed by four authors (LS, SP, MP, MC). If a consensus was not reached, the senior author was consulted (GV). All the retrieved articles were further subdivided into three groups, according to the investigated PMI (short, medium, and long), with the aim of making the data comparable; the records that studied the PMI for up to 7 days were included in the “short PMI” group, those studying PMIs between 8 and 120 days in the “medium PMI” group and the “long PMI” group included all the papers investigating PMIs longer than 120 days. The classification of PMI into these time intervals was chosen to reflect the temporal progression of abiotic and transformative phenomena routinely observed and applied by forensic pathologists in daily practice. In the short PMI, within the first 7 days, consecutive abiotic processes such as livor mortis, rigor mortis, and cooling are observed. In the intermediate PMI, ranging from 7 to 120 days, putrefactive processes predominate, progressing through a series of sequential phases, including chromatic, emphysematous, and liquefactive stages. In the long PMI, beyond 120 days, more advanced transformative phenomena, such as skeletonization, begin to manifest.

The validity assessment of the included manuscript was performed using the ARRIVE guidelines (Animal Research: Reporting of In Vivo Experiments) for animal studies and the STROBE checklist for human studies (see Appendix A for details).

## 3. Results and Discussion

As reported in the PRISMA flow diagram (Figure 1), the combined search on the databases PubMed, Web of Science, and Scopus retrieved 2236 records. Of these, 1084 were duplicates and thus were removed resulting in 1152 articles evaluated by title and abstract. From the latter, 903 were excluded based on criteria A, B, and C, whereas 148 were excluded based on D. The remaining 101 records were analyzed in full-text: 26 were excluded based on criteria A, B, 12 based on E, 9 based on F, and 28 based on G (Figure 1). Twenty-six (26) papers (2.26% of the total) were finally included in the present review [1,2,3,17,19,20,21,22,23,30,34,35,36,38,39,40,41,42,43,44,45,46,47,48,49,50]. All the data extracted from the included records are reported in Table 1.

With regards to the journals publishing the included 26 records, we found that 17 (65.38%) were published in Q1 journals [1,2,3,17,19,20,23,35,36,39,40,41,42,43,44,46,47], 3 (11.53%) in Q2 [21,45,50] and 6 (23.07%) in Q3 [22,30,34,38,48,49]. Some of the retrieved records had more than one quartile score being ranked in different subject areas; in those cases, the best rank was chosen.

Among the 26 included records, 17 papers (65.38%) adopted a proteomic approach [1,2,3,17,19,23,36,38,39,40,41,42,43,45,46,48,49], 9 focused on metabolomics (34.61%) [20,21,22,23,30,34,35,44,47] and only 2 (7.69%) on lipidomics [23,50]. One paper adopted a multi-omics approach focusing on proteomics, metabolomics, and lipidomics [23].

To further correlate the results with the postmortem interval, 15 records focused on a short PMI (57.69%) [3,17,21,22,30,34,35,39,43,44,45,46,47,48,49], 8 on a medium PMI (30.77%) [1,2,20,30,38,41,42,50] and 4 on a long PMI (15.38%) [19,23,36,40].

Interestingly, 11 records (42.30%) analyzed human tissues [1,19,23,30,36,39,40,43,46,48,50], whereas in 18 an experimental approach on animals (69.23%) was described [2,3,17,20,21,22,30,34,35,38,41,42,43,44,45,46,47,49]. Only three of the included records (11.53%) had a double approach, analyzing both animal and human tissues [30,43,46].

The following samples were analyzed: skeletal muscle in nine records (34.61%) [1,3,17,20,30,35,43,44,45], liver in five records (19.23%) [20,38,39,46,49], bones in four records (15.38%) [19,23,36,40], cardiac blood [21,22,34] and leaking fluids [2,41,42] in three records (11.53%), lung [20,39], kidney [20,39] and serum [35,47] in two records (7.69%) and spleen [38], vitreous humor [48], heart [49] and unspecified tissues [50] in one record (3.84%).

To study the correlation between PMI and the retrieved molecules, skeletal muscles, cardiac blood, lung, kidney, liver, serum, vitreous humor, and heart were used for short PMI [3,17,21,22,30,34,35,39,43,44,45,46,47,48,49]; liver, lung, kidney, spleen, skeletal muscles and tissue sampled from upper arm, lower abdomen, and upper thigh were used for medium-term PMI [1,2,20,30,38,41,42,50]; finally, skeletal muscle and bones were used for long PMI [19,23,36,40].

Only 11 (42,31%) of the retrieved records elaborated a predictive model for PMI estimation [19,20,21,22,34,35,44,45,46,47,48].

The ARRIVE guidelines (Animal Research: Reporting of In Vivo Experiments) [51] were applied to the studies in which animals were involved to perform a methodological assessment of the studies’ procedures and design. The first three items (Study design, Sample size, and Inclusion and exclusion criteria) were evaluated positively in the great majority of studies, proving good procedural fairness: most of the studies scored at least 5 out of 6 in the first three items (16/18, 88.88%) [2,3,17,20,21,22,30,34,35,38,41,42,43,44,45,47]. Randomization and Blinding (items 4 and 5) were both performed only in a few studies (2/18, 11.11%) [22,34].

The handling of Experimental animals (item 8) was overall appropriate, providing species-appropriate details (15/18, 83.33%) [2,3,17,20,21,22,30,34,35,41,42,43,45,47,49]; furthermore, few studies provided very accurate details regarding immune systems and previous procedures conducted on the animals (3/18, 16.66%) [21,22,47]. Experimental procedures (item 9) were overall performed in good accordance with the criteria, and only a minority of studies did not satisfy all four of the criteria evaluated on this item (5/18, 27.78%) [3,30,46,47,49]. Results presentation (item 10) was evaluated in accordance with the ARRIVE criteria for most of the studies (16/18, 88.88%) [2,3,17,20,21,22,30,34,35,38,41,42,43,44,46,47]. The results are reported in Appendix A of the Appendix A.

Studies concerning only humans were evaluated using the STROBE checklist [52]. Many items were evaluated as positive in the totality of the articles (8/8, 100%), namely Title and abstract (item 1), Background/rationale (item 2), Objectives (item 3), Study design (item 4), Setting (item 5), Data Sources/measurements (item 8), Outcome data (item 15), Limitations (item 19), Interpretation (item 20), Generalizability (item 21), Funding (item 22). This datum attests to the procedural fairness of these studies. Results are reported in Appendix A.

As demonstrated by the results of the present review, MS-based “omics” sciences have gained increasing importance in the forensic field, proving to be powerful tools for building PMI estimation models. In the following paragraphs, the main results derived from the 26 included records are synthesized and critically discussed according to the PMI (short, intermediate, and long), the type of investigated molecules (proteins, metabolites, lipids) with a molecular insight into the main postmortem time-dependent alterations at the proteomic, metabolomic and lipidomic levels in tissues and fluids (Figure 2).

### 3.1. Short PMI

In the present review, the short postmortem interval was defined as a time elapsed after death of no longer than 7 days.

#### 3.1.1. Proteomics

Most MS-based proteomic research, investigating protein alterations occurring during short PMIs, has focused on muscle tissue analysis of both animal and human origin. In this regard, Marrone et al., analyzing porcine skeletal muscle, identified five main proteins, eEF1A2, eEF2, GPS1, MURC and IPO5, which progressively decreased in the first 24 h after death due to proteolytic enzymes and non-enzymatic processes (i.e., pH and temperature changes), confirming their role as promising biomarkers for short PMI estimation [3]. The time-dependent degradative behavior of eEF1A2 was also investigated and confirmed by Choi et al., who validated its use for short PMI estimation (<96 h) in humans, along with that of GADPH [43]. Further studies on protein degradation kinetics in muscle tissue were conducted by Battistini et al. [17], who highlighted the role of PDLIM7 (PDZ and LIM domain protein 7), TPM1 (Tropomyosin 1) and ATP2A2 (SERCA2) as short PMI biomarkers (<120 h).

In addition to skeletal muscle, other biological substrates have been recently investigated. In particular, Kocsmár et al. explored protein expression patterns in three different human tissue substrates (kidney, liver and lung) finding out tissue-specific protein degradation kinetics, with more delayed breakdown processes occurring in the lung than in other organs [39], thus suggesting an organ-specific susceptibility to autolytic processes.

Although protein degradation patterns are the most frequently observed by researchers, few studies have identified proteins and peptides with different alteration pathways. One of these, conducted on skeletal muscle tissue of pigs, identified four different proteins, SERBP1, COX7B, SOD2 and MAOB, which progressively increased up to 24 h after death, likely due to the activation of hypoxia-induced up-regulation patterns involved in apoptotic, inflammatory and oxidative cellular responses [3]. An analogous incremental kinetics was observed in human vitreous humor samples for Thymosin β4 and two peptide fragments deriving from vimentin and polyubiquitin, which performed well as PMI biomarkers during the time intervals of 15–160 h and 42–160 h after death, respectively. The increases in vimentin and polyubiquitin were related to the cessation of proteasome activity after death, while the up-regulation kinetics affecting Thymosin β4 were attributed to its release by G-actin, following the blockade of different biochemical pathways [48].

Differently, Kwak et al., analyzing heart and liver tissue of rats, described four different protein modification patterns including down-regulation (Tropomyosin, Glutathione synthetase, Cardiac myosin heavy chain 5), up-regulation (Stress-70 protein, Glutathione S-transferase Mu 2, Aldehyde dehydrogenase), down-regulation after up-regulation (Creatine kinase, Aldolase A) and vice versa (Lactate dehydrogenase B, Arginase-1, Hydroxyacylglutathione hydrolase, Nicotinamide adenine dinucleotide), due to the coexistence of both proteolytic mechanisms and cellular up-regulation pathways in response to postmortem hypoxia [49].

Overall, only a few researchers developed predictive models of PMI. Among these, Li et al. investigated, through a MALDI-TOF MS-based approach, the peak distribution of proteins and peptides in muscle and liver tissues of both humans and rats, proposing substrate-specific PMI classification models which showed high recognition capability and cross-validation in distinguishing four different PMI groups within the first 144 h after death (0, 48, 96, 144 h) [45,46].

#### 3.1.2. Metabolomics

All articles focusing on metabolomics and short PMIs were performed on rats; most of them analyzed cardiac blood or skeletal muscle, while a minority explored plasma and serum. Fang et al. analyzed metabolite changes in the cardiac blood of 150 rats at various ambient temperatures, during the first 24 h after death [21], detecting nine metabolites; two of them, Xanthine and Isoleucine, were selected to build an estimation model with an interpolation function, which showed promising results for PMI calculation. Wu et al. and Dai et al. extended the PMI observation to three days, achieving similar results. Wu et al. found up to 55 compounds, which were further used to establish a predictive model with linear regression [34]. Dai et al., instead, used 23 molecules to create an SVR-based PMI predictive model, whose reliability and robustness was further tested, achieving solid results [22].

Working on skeletal muscle, Du et al., profiled the postmortem metabolome up to one week, finding at least 59 compounds which showed a strong correlation with the time since death [44]. Pesko et al. sampled the skeletal muscle of both rats and humans once daily for three days and detected amino acids, decomposition products and other different metabolites [30]; all compounds showed an increasing rate over time due to the prevalence of degradation processes, except for cadaverine, which unexpectedly decreased, likely due to less pronounced microbial activity. Kaszynski et al., retrieved 17 metabolites from the skeletal muscle and 14 from the serum within the first 48 h after death [35]. A prediction model was then established for each retrieved molecule achieving the highest degree of accuracy just for the 24 h after death. Sato et al. managed to isolate 25 metabolites from rats’ plasma, after collecting it for up to 48 h. The compounds were further used for developing a predictive model which showed a good degree of accuracy for predicting PMI [47].

### 3.2. Intermediate PMI

In the present review, the intermediate postmortem interval was defined as a time elapsed after death between 7 and 120 days.

#### 3.2.1. Proteomics

Most studies, investigating protein alterations occurring during intermediate postmortem phases, focused on decomposition fluids of animal origin. Nolan et al., analyzing peptide expression patterns in porcine decomposition fluids under different environmental conditions and over various postmortem time ranges (up to 16 weeks), identified consistent alteration trends for several peptides, deriving from five different proteins including Hemoglobin subunit α and β, Creatine kinase, β-enolase and Lactate dehydrogenase [41,42].

For all that concerns other biological substrates, time-dependent down-regulation patterns were observed for β-actin, SBP2, ENOA and ALDH2 in murine liver samples within the first 240 h after death, highlighting the promising role of these latter three enzymes as biomarkers of the mid-postmortem phase [38].

Novel time-dependent biomarkers were instead identified by Brockbals et al. who proposed the use of multiple peptide ratios, almost all deriving from different human myosin isoforms, for the estimation of decomposition time, up to three months after death. Furthermore, sex- and body mass-specific peptide-ratios were identified, to avoid biases arising from intrinsic biological factors, which proved to affect muscle protein expression due to sexual hormonal differences and variations in microbial activity across BMI categories [1].

#### 3.2.2. Metabolomics

Among the retrieved papers, only two of them investigated the metabolomic profile using mass-spectrometry for medium PMIs. The first authors, Lu et al., focused on four different rat tissues (liver, lung, kidney, and skeletal muscle), sampling them up to 30 days, for this purpose [20]. They found several compounds in the analyzed tissues and a machine learning model was established for each organ for further discriminant analyses. However, through this approach the authors did not obtain reliable results, concluding that this method is not really feasible. Among all the detected compounds, 11 were shared among the four investigated tissues (Nootkatone, Xanthine, Hypoxanthine, Azelaic Acid, Acetophenone, *N*-Acetylhistamine, *N*-Acetylvaline, 3-Phenyllactic Acid, Indole-3-Lactic Acid, *N*-Acetyl-l-Phenylalanine, and *N*-Acetyl-DL-Tryphtophan). Interestingly, the multi-organ stacking model outperformed the previous mentioned single-organ approach.

Pesko et al. analyzed the muscle tissue sampled from human frozen cadavers for up to 19 days postmortem [30]. The following metabolites of interest were found: Skatole, Xanthine, *N*-Acetylneuraminate, 1-Methylnicotinamide, Choline Phosphate, Uracil, Tyrosine, Threonine, and Lysine. Overall, these compounds showed a general increasing trend in the tested tissue due to protein degradation patterns and microbial decomposition activity of macromolecules; in particular, among amino acids, Lysine showed the best promising results and *N*-Acetylneuraminate the highest rate of increase over time.

#### 3.2.3. Lipidomics

Ueland et al. explored saturated and unsaturated fatty acids, dicarboxylic acids, and sterols, sampling human tissue from the upper arm, lower abdomen/torso region and the buttocks/upper thigh, from one fresh and one frozen human cadaver for up to 69 days [50]. Among all the compounds, Stearic and Palmitic acid showed increased relative abundance during specific decomposition stages, suggesting their utility for estimating PMI. About unsaturated fatty acids (e.g., Oleic and Linoleic Acids) the authors noticed that their changes were useful to mark the transition between decomposition stages. In regards of other molecules, sterols were found to be generally stable; though this was in line with previous archeological studies, their lack of specificity and consistent stability make them less reliable for PMI estimation. Finally, concerning dicarboxylic acids, no significant statistical differences between donors or regions were found, limiting their forensic utility for PMI evaluation.

### 3.3. Long PMI

In the present review, the long postmortem interval was defined as a time elapsed after death longer than 120 days.

#### 3.3.1. Proteomics

Whilst most studies have focused on protein alterations occurring during the early and mid-postmortem phases in various types of biological substrates, only a few studies have investigated the role of these biochemical changes during longer PMIs, all focusing on bone tissue analysis. In fact, bones proved to be a useful matrix for exploring the latter phases of PMI, especially those extending over years or decades.

In this regard Mickleburgh et al. observed time-dependent protein degradation kinetics on both fresh (PMI: 2–10 days) and skeletonized human bones, showing that collagen and mineral-related proteins as COBA2, CO3A1, PGS2 and MGP are suitable candidates for longer PMIs, whereas blood proteins as CO3, CO9 and TTHY are valid biomarkers for shorter PMIs, due to their greater susceptibility to bioerosion and diagenesis [40]. Their findings also suggested that taphonomy and intrinsic biological factors play a significant role in bone protein survival, recommending consideration of these variables, together with inter-skeletal differences in protein diversity and abundance, for a more reliable PMI estimation. Breakdown patterns have also been described for Collagen protein type I, whose peptide concentrations showed high discriminatory ability in differentiating PMIs shorter than 20 years from PMIs longer than 20 years [19].

Bonicelli et al. confirmed down-regulation kinetic trends with increasing PMI (1–37 years) for most bone-related proteins including B2MG (β-2-microglobulin), CSPG2 (versican core protein), G3P (glyceraldehyde-3-phosphate dehydrogenase), IGL1 (immunoglobulin lambda-like polypeptide 1), KNG1 (kininogen-1), PCOC1 (procollagen C-endopeptidase enhancer 1), PGBM (basement membrane-specific heparan sulfate proteoglycan core protein) and RCN3 (reticulocalbin-3), with the only exception of ANT3 (Antithrombin III) and CHAD (Chondroadherin), which showed a weak up-regulation pattern [36]. Moreover, due to their greater longevity and relative stability, non-collagenous proteins (NCPs) proved to be better biomarkers of long PMIs (CSPG2 showed the strongest correlation with PMI: r_s_ = −0.68 with *p* < 0.0001) than collagen and collagen-binding proteins (only one collagen-binding protein, PCOC1, showed a significant correlation with PMI: r_s_ = −0.45 with *p* < 0.05), confirming their usefulness also in archaeological contexts. Finally, in another work, the same authors proposed two histone proteins (H2A1H, H4), Haemoglobin subunit alpha (HBA), actin (ACTB) and vimentin (VIME) as the best biomarkers to discriminate between the “fresh” bones and the “skeletonized” ones (r > 0.9) [23].

#### 3.3.2. Metabolomics

Bonicelli et al., analyzing bone samples at four selected long PMIs (e.g., from 219 to 872 days), found several key metabolite compounds including amino acids, nucleotides, fatty acids and relative by-products, organic acids, CAPSO and Sedanolide [23].

Molecules such as Palmitoyl Ethanolamide, Ethyl Palmitolate, *N*,*N*-diethylethanolamine, sedanolide, 12-Aminododecanoic acid and Acetamide increased over time, suggesting time-dependent progressive decomposition trends. Instead, other metabolites such as Hypoxanthine, Creatine and Taurine showed a consistent decline over time, with a significant drop within 219 days, suggesting that earlier PMI phases are driven by compound decomposition, then followed by metabolic arrest.

Among all compounds, Acetamide showed the best positive correlation with PMI (r > 0.9), resulting in the most reliable and promising biomarker.

#### 3.3.3. Lipidomics

In the same paper [23], through a lipidomic approach Bonicelli et al. isolated three Lysophosphatidylcholines (LPCs), one Phosphatidylcholine (PC) and one Phosphatidylinositol (PI), all showing lower intensities in the decomposed samples than in the “fresh” ones. The time-dependent decline of these lipids aligns with postmortem degradation, making them useful markers for estimating PMIs, particularly in differentiating between fresh and decomposed bone states. Although these molecules were proposed by the authors as promising long PMI biomarkers, their alteration trends proved to be influenced by environmental factors and interindividual variability, thus highlighting the need for further validation.

### 3.4. Postmortem Up and Down-Regulation Patterns of Biomolecules. A Molecular Insight

As shown in Table 2, and in more detail in Appendix A, recent mass-spectrometric “omic” investigations, above all with a proteo-metabolomic untargeted approach, unraveled interesting molecular postmortem regulation patterns. Time-dependent degradation kinetics have been reported for several proteins and involve different postmortem biochemical mechanisms, including hypoxia-induced autolysis (consisting of cell disruption and subsequent extracellular release of lysosomal enzymes), non-enzymatic decomposition processes (pH and body temperature changes), and proteolytic activities of endogenous and/or exogenous microorganisms colonizing corpses [3,24,49,53,54,55,56,57,58,59,60,61,62]. The above time-dependent breakdown rates are generally protein-specific, due to different susceptibilities to enzymatic degradation [3,63]. Different amino acid sequences and post-translational alterations influence molecular spatial structures and functions, and thus inter-molecular interactions, leading to great variability in protease cleavage sites, with more resistant proteins increasing their relative abundance within the first hours after death [25]. This time-dependent up-regulation trend can sometimes be traced back to the activation of specific signal–transduction pathways triggered by hypoxic alterations occurring after death. Systemic oxygen deficiency related to circulatory cessation disrupts cell redox homeostasis through the production of reactive oxygen species (ROS), leading to oxidative stress-induced cell death and molecular damage [64,65]. Therefore, the significant increase in mitochondrial enzymes such as COX7B and MAOB, observed by Marrone et al. [3] in muscle tissue during short PMIs, can be considered a direct expression of the hypoxia-induced oxidative status affecting the postmortem cellular microenvironment. The anoxic insult is also well-known to trigger the activation of cellular self-defense reactions and protection mechanisms, aimed at repairing stress-induced macromolecular damage and re-establishing biological homeostasis. Interestingly, most of the proteins increasing their relative abundance in the early PMIs, including Thymosinß4, SOD2, Stress-70 protein, Glutathione S-transferase Mu 2 and Aldehyde dehydrogenase, share the same anti-oxidant, anti-apoptotic, anti-inflammatory and/or tissue-repairing properties [48,49]. Their increase shortly after death can therefore be considered an expression of defensive and adaptive mechanisms implemented by cells in response to pro-oxidative, pro-inflammatory and pro-apoptotic alterations typically affecting the postmortem catabolic microenvironment.

As observed for proteomics, metabolomics alteration patterns are driven by a combination of multiple biochemical mechanisms including autolysis, microbial metabolism and shifts in cellular energy homeostasis [66,67]. Postmortem systemic hypoxia results in cellular energy deficiency which leads to the disruption of biomolecular synthesis and to the increase of biomolecular catabolic processes, with subsequently progressive increase of amino acids and nucleotides, as observed by many researchers [20,34,47]. Furthermore, a progressive accumulation of biogenic amines including putrescine, indole and skatole has been observed, resulting from bacteria-induced breakdown of amino acids. The gradual reduction of pyruvate and the simultaneous increase in lactic acid concentrations observed by Dai et al. [22] and Wu et al. [34] can be explained by postmortem cellular shift from aerobic to anaerobic metabolism, which leads to the arrest of citric acid cycle (CAC) and to subsequent activation of fermentation pathway.

Two main types of alteration kinetics have been observed for lipids. In particular, hydrophobic acids as palmitic, stearic, oleic and linoleic acids were found to progressively increase after death (Ueland et al.) [50]. According to other studies, this increase can be explained by progressive lipase-mediated hydrolysis, which breaks down triglycerides in free fatty acids and glycerol [68,69]. Interestingly, stearic and palmitic acids showed concentration peaks during the bloat stage of decomposition. This trend can be related to the intense microbial activity affecting this specific postmortem phase and the relative higher stability of saturated fatty acids compared to unsaturated ones [70].

Unfortunately, the majority of the included records (Table 1) analyzed animal biological substrates, raising concerns about the applicability of these findings on humans. Although protein alteration patterns are common to both animals and humans, an interesting study has detected, by comparing the same rat and human biological matrices, different protein/peptide peaks during the same postmortem time range [46]. These differences may reflect the distinct biological microenvironment characterizing different species, due to diverse microbial actions and enzymatic activities. Therefore, species-specific PMI predictive models should be established to account for these biological differences and to obtain a more reliable PMI estimation. Moreover, Kocsmár et al., analyzing protein alteration patterns in different tissues, observed an organ-specific degradation kinetic [39]. This inter-organ variability can be explained by the different susceptibility of tissues to autolysis and decomposition processes, which is closely related to the visceral storage of self-digestive enzymes as well as to the organ content of “soft and hard tissues”, with collagen and keratin conferring greater resistance to enzymatic degradation [24]. Indeed, bone tissue proved to be the most suitable biological substrates for the estimation of longer PMIs, when soft tissues are unlikely to be preserved. This may be related to the protective role of hydroxyapatite, which enhances a longer survival of collagen and mineral-binding proteins (CO3A1, COBA2, MGP) than plasma/blood proteins (CO3, CO9, TTHY) protecting the former from microbial bioerosion and diagenesis [19,23,40,71]. Non-collagenous proteins (NCPs) with high affinity to bone mineral matrix proved to survive even longer than collagen-related proteins, due to their better resistance to breakdown processes, suggesting their promising role as long PMI biomarkers [36]. The same inter-organ variability affects also metabolomics approaches. In particular, Lu et al., analyzing metabolomic profiles of four different rat tissues (skeletal muscle, liver, lung and kidney) identified different organ sensitivities to estimate PMI [20]. These inter-organ variations in decomposition patterns may be related to both the heterogeneity of microbial proliferation colonizing different tissues and diverse organ susceptibility to autolysis and decomposition processes. Therefore, the application of a multi-organ predictive model could enhance accuracy and precision of PMI estimation compared to a single organ-based model [23]. Furthermore, as degradation kinetics proved to be molecule specific, with different compounds presenting distinct temporal sequences of decay, PMI estimation based on the analysis of one or few biomarkers should be avoided. Differently, combining patterns from multiple markers is recommended for a more accurate PMI estimation.

Finally, it is crucial to highlight that postmortem biochemical alterations are also significantly influenced by a wide range of factors, which should be considered when estimating the PMI. These include intrinsic factors (sex, age, BMI, cause of death and ante-mortem comorbidities), and extrinsic factors such as temperature, burial conditions, action of microorganisms, insects and scavengers [63,72,73,74,75,76,77,78,79]. In particular, Brockbals et al. identified sex-specific PMI biomarkers, highlighting a sexual dimorphism in muscle protein expression, linked to hormonal variations. BMI-specific biomarkers were also identified, underscoring the influence of body mass on the postmortem microbiota and, consequently, on decomposition patterns [1]. Nonan et al. emphasized the influence of temperature on PMI, with higher temperatures accelerating protein degradation kinetics [2] whereas Bonicelli et al. investigated the impact of burial conditions (entombment vs. inhumation) on postmortem survival of non-collagenous bone proteins (NCPs), showing better preservation of entombed bones than inhumed ones, related to the lower influence on the former of the exogenous microbial activity [36].

The role of xenobiotics is also noteworthy, as several studies demonstrated that the presence of drugs or toxins can significantly alter corpse decomposition processes by affecting microbial activity as well as colonization patterns and life cycles of cadaveric entomofauna [80,81,82,83]. However, only one paper included in the present review specifically examined the impact of toxicants, revealing that DDVP (dichlorvos) exposure in rats influences postmortem metabolic profiling, leading to a reduction in energy-related metabolites such as valine, isoleucine, and pyruvate due to an increase of perimortem muscle activity [22].

## 4. Conclusions

Although MS-based “omics” approaches proved great potential, several significant challenges remain. As previously discussed, postmortem biochemical processes proved to be strongly influenced by a broad spectrum of factors, which should not be overlooked when estimating the PMI through the application of “omics” approaches.

Furthermore, the heterogeneity of the studies included in the present review does not allow definitive conclusions to be drawn. Great variations in sample sizes and types, experimental conditions, analytical methodologies and statistical techniques applied in each animal or human investigation here included, make direct comparison among the various findings very difficult, introducing pitfalls and interpretation bias.

To address these limitations, several potential solutions may be proposed.

First, it is essential to establish predictive models for PMI that incorporate the aforementioned influencing factors and variables (i.e., temperature, sex, BMI, drug exposure, burial conditions), thereby minimizing bias and enhancing the accuracy and reliability of estimations.

Furthermore, given the time-dependent alteration patterns of biomolecules, a combined analysis of multiple “omics” panels is recommended to improve the precision and accuracy of PMI estimation. In this context, the standardization of extraction protocols and analytical procedures, coupled with the application of rigorous statistical methods, is crucial for advancing omics-based research in PMI estimation.

Additionally, a comparative analysis between omics-based approaches and conventional gold standard methods (such as K + levels in the vitreous humor) routinely used in forensic casework should be conducted to validate the proposed biomarkers.

Overall, MS-based omics research has led to a breakthrough in the forensic field, paving the way for the discovery of novel promising PMI biomarkers for more accurate and reliable PMI estimations, above all for intermediate and long PMIs, which remain a challenge for forensic pathologists and entomologists. More importantly, further research aiming at elucidating the molecular, time-dependent and tissue-specific mechanisms of postmortem modifications might impact not only forensic research and practice but also transplantation medicine, where understanding the preservation of organs and tissues during warm and cold ischemia is crucial for improving organ viability and enhancing transplantation outcomes.

## Figures and Tables

**Figure 1 ijms-26-01034-f001:**
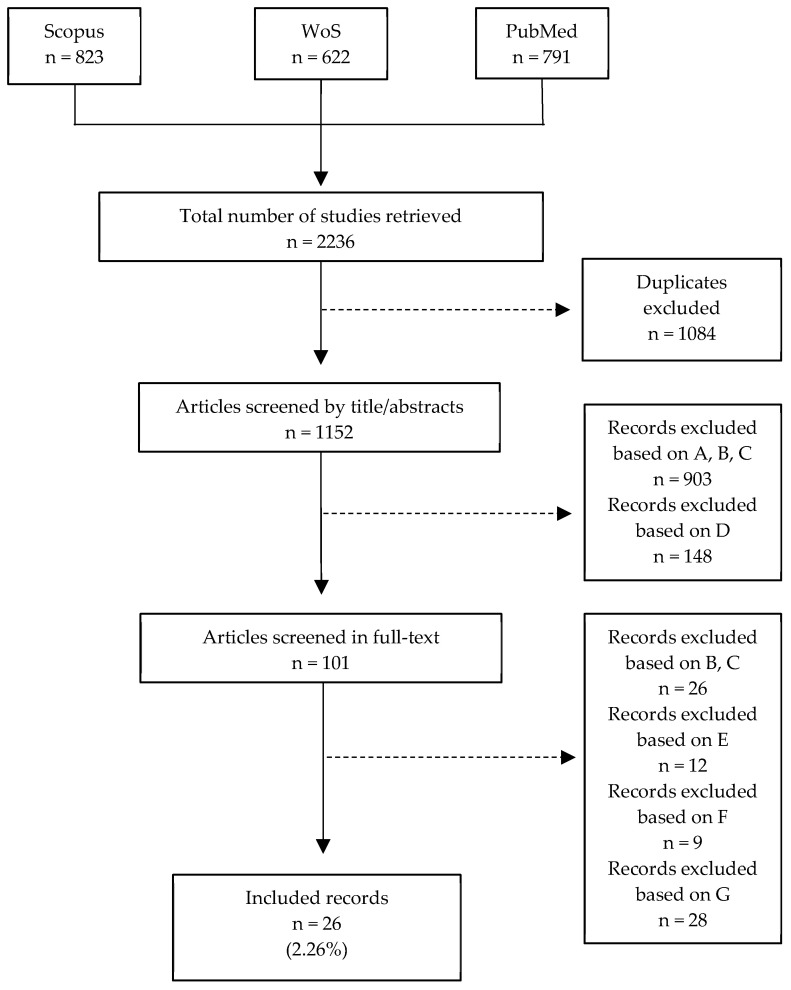
PRISMA flow diagram.

**Figure 2 ijms-26-01034-f002:**
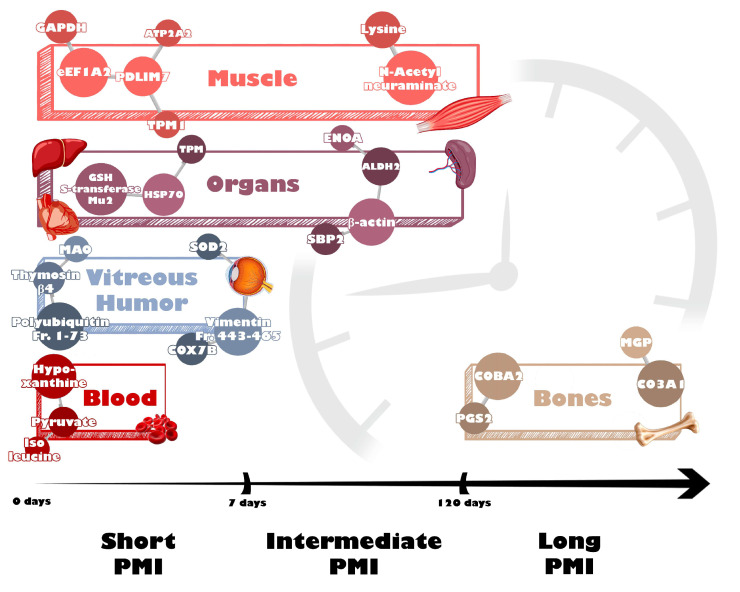
Main tissues and macromolecules showing time-dependent changes in short, intermediate, and long PMIs according to the 26 included records.

**Table 1 ijms-26-01034-t001:** Data extracted from the 26 included records: first author, year and journal of publication, best JCR quartile (2024), sample size and type (A for animal, H for human), time interval after death explored (hours or days), analytical technique used, statistical approach, main biomarkers, presence of a PMI predictive model, diagnostic efficiency, and error interval/accuracy of the selected markers.

Study, Year and JCR Rank [Citation]	Sample and Time Interval	Technique and Approach	Statistical Approach	Biomarkers	Predictive Model
					Efficiency
					Error Interval-Accuracy
Zhang W. et al., 2024, Q3 [38]	A (42 mice) Liver and spleen 0–240 h	MALDI-TOF/TOF Proteomic	Student *t*-test (*p* < 0.05)	6 proteins: SBP2, ENOA, ALDH2, 3HAO, TPIS, CATA	No No No
Marrone A. et al., 2023, Q1 [3]	A (3 pigs) Skeletal muscle 0–24 h	LC-MS/MS Proteomic	Student *t*-test (*p* < 0.01, S0 = 0.2)	9 proteins: eEF1A2, eEF2, GPS1, MURC, IPO5; SERBP1, COX7B, SOD2, MAO	No No No
Brockbals L. et al., 2023, Q1 [1]	H (9) Skeletal muscle 0–120 days	LC-MS/MS Proteomic	Coefficient of determination (r^2^ ≥ 0.5)	12 peptides ratios (main proteins of origin Myosin-2 and Myosin-7)	No No No
Battistini A. et al., 2023, Q1 [17]	A (3 pigs) Skeletal muscle 0–120 h	LC-ESI-MS/MS Proteomic	ANOVA + Tukey test (n = 3, *p* < 0.05)	22 proteins: AK1, BAG3, MYOZ3, MYOZ1, ACY1, PDLIM7, PDLIM3, PRDX2, LOC100156324, SYNPO2, DHRS7C, RAB10, RTN2, SRL, TPM1, ATP5MF, ATP2A2, TNNC1, ATP5PB, ATP5MG, PDLIM5, SLC25A4	No No No
Fang S. et al., 2023, Q2 [21]	A (150 rats) Cardiac blood 0–24 h	GC-MS Metabolomic	PCA, PLS, VIP > 1.0, Kruskal–Wallis test (*p* < 0.001)	4 amino acids: isoleucine, alanine, proline, valine. Other molecules: glycerol, glycerol phosphate, xanthine, and hypoxanthine	Yes Yes Yes
Kocsmár E. et al., 2023, Q1 [39]	H (3 to 4) Lung, kidney, liver 6–96 h	LC-MS/MS Proteomic	Benjamini–Hochberg, PCA	18 proteins for each investigated organ: liver, lung, and kidney	No No No
Lu X. et al., 2023, Q1 [20]	A (140 rats) Skeletal muscle, liver, lung, kidney 0–30 days	UPLC-HRMS Metabolomic	PCA, OPLS-DA, permutation test, R^2^ ≈ 1, Q^2^ ≥ 0.5, Mann–Whitney U test (*p* < 0.01)	11 main metabolites: Nootkatone, Xanthine, Hypoxanthine, Azelaic Acid, Acetophenone, *N*-Acetylhistamine, *N*-Acetylvaline, 3-Phenyllactic Acid, Indole-3-Lactic Acid, *N*-Acetyl-L-Phenylalanine, and *N*-Acetyl-DL-tryphtophan	Yes Yes Yes
Bonicelli A. et al., 2022, Q1 [23]	H (4) Bone 2–872 days	LC-MS Proteomic Metabolomic Lipidomic	PCA, Kruskal–Wallis, Dunn’s test, Holm’s correction, PLS-DA, regression PLS pairwise	18 main molecules (most relevant H2A1H, H4, VIME, ACTB, HBA, Palmitoyl ethanolamide, *N*,*N*–Diethylethanolamine, Creatine, Hypoxanthine, Creatinine, CAPSO, Sedanolide, Taurine, Uracil, Ethyl palmitoleate, 12-Aminododecanoic acid, 12-hydroxydodecanoic acid, d-Neopterin)	No No Yes
Bonicelli A. et al., 2022, Q1 [36]	H (14) Bone 1–37 years	LC-MS/MS Proteomic	Student *t*-test, Robust empirical Bayes regression, Spearman’s rank correlation (*p* < 0.05)	90 proteins (most relevant ALBU, ASPN, CLC11, FETUA, FMOD, MIME, NUCB1, B2MG, KNG1, PCOC1, PGMB, CSPG2, G3P, IGL1, RCN3)	No No No
Mickleburgh H. et al., 2021, Q1 [40]	H (4) Bone 2–872 days	LC-MS/MS Proteomic	Scaffold local FDR, Mascot evaluation, ANOVA, post hoc mean pairwise, Wilcoxon rank sum test, and Kruskal–Wallis	7 main proteins: CO3A1, CO9, COBA2, MGP, PGS2, TTHY, CO3	No No No
Pesko B. et al., 2020, Q3 [30]	A (8 rats) Skeletal muscle 0–72 h H (6) Skeletal muscle 3–19 days	LC-MS Metabolomic	Retention time prediction algorithm	20 main metabolites in rats: Methionine, Tryptophane, Leucine, aspartate, Phenylalanine, Valine, Lysine, Tyrosine, Histidine, Threonine, Arginine, Cadaverine, Putresceine, Skatole, Indole, Xanthine, N-acetylneuraminate, Uracil, Choline Phosphate, 1-methylnicotinamide 9 main metabolites in humans: Tyrosine, Threonine, Lysine, Skatole, Xanthine, *N*-acetylneuraminate, 1-methylnicotinamide, Choline phosphate, Uracil.	No No No
Nolan A. et al., 2020, Q1 [41]	A (16 pigs) Fluid leaking from the body <34 days	HPLC-MS Proteomic	Multivariate analysis, PCA, Student *t*-test	29 peptides resulting from the degradation of HBA, HBB, CK, ENO3, LDH	No No No
Nolan A. et al., 2019, Q1 [42]	A (16 pigs) Fluid leaking from the body <34 days	HPLC-MS Proteomic	VennDiagram package	142 peptides mainly resulting from the degradation of HBA, HBB, CK, ENO3	No No No
Nolan A. et al., 2019, Q1 [2]	A (4 pigs) Fluid leaking from the body <4 weeks	HPLC-TOF Proteomic	VennDiagram package	27 peptides resulting from degradation of CK, ENO3, PK, HBA, HBB	No No No
Choi K. et al., 2019, Q1 [43]	A (25 Rats, 10 Mice) Skeletal muscle 0–96 h H (3) Skeletal muscle 0–96 h	LC-MS/MS Proteomic	Spearman correlation, Kolmogorov-Smirnov tests, ANOVA, Tukey’s post hoc multiple comparisons tests, Kruskal–Wallis test, Bonferroni correction method (*p* ≤ 0.05, *p* ≤ 0.001 highly significant)	2 proteins: eEF1A2, GAPDH	No No No
Dai X. et al., 2019, Q3 [22]	A (36 Rats) Cardiac blood 0–72 h	GC-MS Metabolomic	PCA, PLS, VIP > 1.0, Kruskal–Wallis test (*p* < 0.05)	23 main metabolites: valine, phenylalanine, leucine, xanthine, ribitol, isoleucine, 3-aminoisobutyric acid, threonine, mannitol, creatinine, pantothenate, pyroglutamate, xylose, hypoxanthine, linoleic acid, palmitic acid, pentitol, malic acid, pyruvate, proline, methionine, glutamate, uracil	Yes Yes Yes
Du T. et al., 2018, Q1 [44]	A (60 Rats) Skeletal muscle 3–168 h	LC-MS Metabolomic	PCA, PLS-DA, OPLS-DA, VIP > 1.5, Student’s *t*-test (*p* < 0.05)	14 main metabolites: cytidine, isomaltose, UDP-N-acetylglucosamine, inosine 5-monophosphate, uridine 5-monophosphate, guanosine 5-monophosphate, nicotinamide, 3-o-metylguanosine, reduced nicotinamide adenine dinucleotide, beta-nicotinamide, d-ribonucleotide, d-alanyl-d-alanine, glycerol 3-phosphate, n6-acetyl-l-lysine	Yes No No
Wu Z. et al., 2018, Q3 [34]	A (84 Rats) Cardiac blood 0–72 h	GC-MS Metabolomic	PCA, OSC-PLS, VIP > 1.2, Kruskal–Wallis test (*p* < 0.05), Pearson correlation coefficient	55 metabolites: mainly organic acids, amino acids, carbohydrates, lipids, and others	Yes Yes Yes
Pérez-Martìnez C. et al., 2017, Q1 [19]	H (80) Bone More o less than 20 years	HPLC-MS/MS Proteomic	Spearman’s coefficient, Kruskal–Wallis test (*p* ≤ 0.05)	8 nitrogenous bases (adenine, guanine, purines, cytosine, thymine, pyrimidines, hypoxanthine and xanthine), and several Collagen Type I peptides	Yes Yes Yes
Li C. et al., 2017, Q2 [45]	A (4 Rats) Skeletal muscle 0–144 h	MALDI-TOF MS Proteomic	PCA, Wilcoxon test (*p* < 0.05)	Skeletal muscle proteins (not specifically reported by the authors)	Yes Yes Yes
Li C. et al., 2017, Q1 [46]	A (36 Rats) Liver 0–144 h H (4) Liver 6–168 h	MALDI-TOF MS Proteomic	PCA, GA, SNN, QC algorithms	4 main proteins in rats: LOC102151723, basic proline-rich protein-like, olfactory receptor 2G3-like, interferon omega 5 precursor. 3 main protesins in humans: Rho GTPase-activating protein 24, Amine oxidase, Small vasohibin-binding protein	Yes Yes Yes
Kaszynski R. et al., 2016, Q1 [35]	A (52 Mice) Serum and skeletal muscle 0–48 h	GC-MS Metabolomic	Pearson’s correlation coefficient (*p* < 0.05), PCA	17 main metabolites in muscle: Nicotinamide, Hypoxantine, *N*-formylglycine, Medo-erythritol, Citrulline, Valine, Lysine, 2-Aminoethanol, *N*-methylethanolamine, isoleucine, threonine, galactosamine, S-benzyl-l-cysteine, Methionine, Pyroglutamic acid, Ribitol, Xylitol. 15 main metabolites in serum: Hydroxybutyrate, glucarate, xylitol, glycerol-2-phosphate, ribitol, rhamnose, 2-aminoethanole, pantothenate, b-alanine, ribose, tryptophane, lysine, glucosamine, citric acid, isocitric acid.	Yes Yes Yes
Sato T. et al., 2015, Q1 [47]	A (36 Rats) Serum 0–48 h	GC-MS/MS Metabolomic	PCA, PLS regression model, Kruskal–Wallis test, VIP > 1.2	18 amino acids, 5 sugars, 1 carboxylic acid, 1 phosphate	Yes Yes Yes
Boroumand M. et al., 2023, Q3 [48]	H (7) Vitreous humor 3–160 h	High resolution HPLC-ESI-MS, MS/MS Proteomic	Correlation analysis, Pearson correlation coefficient	7 proteins and 35 peptide fragments	Yes No No
Kwak J. et al., 2016, Q3 [49]	A (3 Rats) Liver and heart 0–48 h	HPLC-MS Proteomic	Cross-correlation, delta correlation	9 main proteins: Tropomyosin, Cardiac myosin heavy chain 5, stress-70, Glutathione synthetase, Mu 2, Glutathione S-transferase alpha, Creatine kinase, Enolase, Aldehyde dehydrogenase.	No No No
Ueland M. et al., 2021, Q2 [50]	H (2) Unspecified tissue from upper arm, lower abdomen, upper thigh 0–69 days	GC–MS/MS Lipidomic	PCA, Two-way ANOVA with post hoc tests, Shapiro–Wilk tests	22 fatty acids (saturated, unsaturated, and dicarboxylic acids) and 11 sterol analytes	No No No

Legend: A = animal; DA = discriminant analysis; FDR = false discovery rate; GA = genetic algorithm; GC = gas-chromatography; h = hours; H = human; HPLC = high-performance liquid chromatography; LC = liquid chromatography; MALDI = matrix-assisted laser desorption; MS = mass spectrometry; OPLS = orthogonal projections to latent structures; PCA = principal component analysis; PLS = partial least squares; QC = quality control; SNN = spiking neural networks; TOF = time-of-flight; UPLC = ultra-performance liquid chromatography; VIP = variable importance of projection. For other abbreviations, please see the “Abbreviations” section.

**Table 2 ijms-26-01034-t002:** Main down- and/or up-regulated molecules during different PMIs (short, intermediate and long).

		Short PMI	Intermediate PMI	Long PMI
Down-regulation pattern	**↓**	ATP2A2 (SERCA2),Cadaverine,Cardiac myosin heavy chain 5,eEF1A2,eEF2,GAPDH,Glutathione synthetase,GPS1,IPO5,MURC,PDLIM7,Pyruvate,TPM1 Tropomyosin.	ALDH2Azelaic Acid,ß-actinß-enolase,CK,ENOA,HBA,HBB,Hypoxanthine,LDH,Nootkatone,SBP2.	ACTB,B2MG,CAPSO,CO3,CO3A1,CO9,COBA2,Creatine,CSPG2,G3P,H2A1H,H4,HBA,Hypoxanthine,IGL1,KNG1,LPC(17:0) + HCOO,LPC(18:0) + HCOO,LPC(19:0) + HCOO,MGP,PC(16:1e_20:4) + HCOO,PCOC1,PI(18:0_20:4)-H,PGBM,PGS2,RCN3Taurine,TTHY,VIME.
Up-regulation pattern	**↑**	Aldehyde dehydrogenase,COX7B,Glutathione S-transferase Mu2,Indole,Isoleucine,Lactic acid,MAOB,Oleic acid,Palmitic acid,Polyubiquitin Fr. 1–73,PutresceineSERBP1,SOD2,Stearic acid,Stress-70 protein,Thymosin ß4,Vimentin Fr. 443–465,Xanthine.	1-Methylnicotinamide,3-Phenyllactic Acid,Acetophenone,Choline Phosphate,Indole,Indole-3-Lactic Acid,Linoleic acidLysine,*N*-Acetyl-DL-tryphtophan,*N*-Acetylhistamine,*N*-Acetyl-l-Phenylalanine,*N*-Acetylneuraminate,*N*-Acetylvaline,Oleic acid,Palmitic acid,Skatole,Stearic acid,Threonine,Tyrosine,Uracil,Xanthine.	12-Aminododecanoic acid,Acetamide,ANT3,CHAD,Ethyl palmitolate,*N*,*N*-Diethylethanolamine, Palmitoyl ethanolamide,Sedanolide.
Up-regulation followed by down-regulation	**↑↓**	Aldolase A,Creatine kinase.		
Down-regulation followed by up-regulation	**↓↑**	Arginase-1,Hydroxyacylglutathione hydrolase,Lactate dehydrogenase B,Nicotinamide adenine dinucleotide.

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
