# Peer review of "“Omics” and Postmortem Interval Estimation: A Systematic Review"

_ijms, 2025, doi:10.3390/ijms26031034_

Round 1

Reviewer 1 Report

Comments and Suggestions for Authors

This review is a really interesting work for professionals working in the field of forensic pathology and toxicology. The determination of  postmortem interval (PMI) is a crucial issue, especially in cases with a cadaver in a crime scene, where reliable and accurate estimation of the PMI is important for the determination of the time of the crime.

There are only few reviews published that investigate the estimation of PMI using an “omic” approach.

This work is scientifically sound and the results –discussion section is based on the data exported from the 26 articles included in this review.

References are well placed and include related articles and finally the paper is well written.

Only few observations:

1.       Figure 2 should be bigger since it is difficult to read the small letters in the texts provided.

2.       In my point of view paragraph starting in line 505-511 is really important, since both intrinsic and extrinsic factors in deed affect post mortem biochemical processes and for this reason these factors should be taken into account. This paragraph should be expanded and present data from these articles for example: ratio male /female , temperature differences e.t.c

3.       Also, is the results of the toxicological analysis presented in the articles with human cadavers and does the consumption of drugs or drugs of abuse  affect "perimortem"  the up or down regulation of the molecules studied in these works? A paragraph could be included regarding the consumption of drugs and the effect it may have in the estimation of PMI.

Based on the above I would consider acceptance of this review after minor revisions

Reviewer 2 Report

Comments and Suggestions for Authors

1. The description of the journal classification of the included articles (Page 2, Lines 208–212) could be moved earlier in the text to improve the logical flow of the paragraph.

2. The paper lists the main results of the included studies; however, the logical connections between these results are weak.

3. Adding a brief discussion of the possible mechanisms behind the experimental results, as mentioned in the original studies, could enhance the scientific significance of the article.

4. The rationale for defining short, intermediate, and long PMI time ranges in this paper could be briefly introduced.

5. Please verify the accuracy of protein names, such as eEF1A2 in Table 1, which is referred to as EeF1A2 and Eef1a2 in the main text (Page 10, Line 257). Ensure consistent and standard nomenclature throughout the article.

6. Vague descriptions such as “better candidates” or “most promising biomarker” should be minimized. Supporting data or experimental results should be provided to justify terms like “better” or “most.”

7. The logical structure of the Conclusion section is somewhat loose, with challenges, solutions, and future perspectives intertwined. It is recommended to separate this section into three distinct parts: (i) current challenges; (ii) proposed solutions; and (iii) future research directions.

Reviewer 3 Report

Comments and Suggestions for Authors

The paper is certainly interesting and innovative. The review seems to have followed all the PRISMA criteria and it is scientifically reliable. However, I do have a couple of suggestions for minor revisions.

Line 17 and 72: define "omics" sciences;

Introduction: the quality of the English language should be improved;

Table1: add a list of all abbreviations used at the end of the paper, e.g. AK1

Line 257 and 259: check all abbreviations used, they must all be spelt the same, i.e. EeF1A2 and Eef1a2.

Section 3.4: it could be useful for the readers to add a table of all the up-regulated and down-regulated biomolecules.

Conclusion: At the end of any review, I would like to read an in-depth analysis by the authors. You could use all the data you have extrapolated to explore the biological pathways that you think are more promising and explain why.

Comments on the Quality of English Language

I suggest you check the quality of the English language, it seems to be an Italian text translated into English. The sentences are too long and full of unusual words. With the help of a native English speaker, you need to check the whole text carefully, please.
